# Synovium-Synovial Fluid Axis in Osteoarthritis Pathology: A Key Regulator of the Cartilage Degradation Process

**DOI:** 10.3390/genes12070989

**Published:** 2021-06-29

**Authors:** Dhanashri Ingale, Priya Kulkarni, Ali Electricwala, Alpana Moghe, Sara Kamyab, Suresh Jagtap, Aare Martson, Sulev Koks, Abhay Harsulkar

**Affiliations:** 1Department of Cell and Molecular Biology, Interactive Research School for Health Affairs (IRSHA), Bharati Vidyapeeth University, Pune 411043, India; ingaledhanashri91@gmail.com; 2Department of Pathophysiology, Biomedicine and Translational Medicine, University of Tartu, Ravila 19, 50411 Tartu, Estonia; 3Department of Traumatology and Orthopedics, Tartu University Hospital, University of Tartu, L Puusepa 8, 51014 Tartu, Estonia; Aare.Martson@kliinikum.ee; 4Electricwala Hospital, A 4/1, Pleasant Park, Fatima Nagar, Wanowrie, Pune 411013, India; ali.electricwala@gmail.com; 5Department of Cell and Molecular Biology, Rajiv Gandhi Institute of IT and Biotechnology, Bharati Vidyapeeth University, Pune 411043, India; alpanasm@gmail.com (A.M.); sara.kamyab@gmail.com (S.K.); 6Herbal Medicine, Interactive Research School for Health Affairs (IRSHA), Bharati Vidyapeeth University, Pune 411043, India; chiritatml@rediffmail.com; 7Clinic of Traumatology and Orthopaedics, Tartu University Hospital, University of Tartu, L Puusepa 8, 51014 Tartu, Estonia; 8Centre for Molecular Medicine and Innovative Therapeutics, Murdoch University, Murdoch, WA 6150, Australia; sulev.koks@perron.uwa.edu.au; 9Perron Institute for Neurological and Translational Science, Nedlands, WA 6009, Australia; 10Department of Pharmaceutical Biotechnology, Poona College of Pharmacy, Bharati Vidyapeeth University, Erandwane, Pune 411038, India

**Keywords:** FALGPA, inflammatory mediators, Kellgren–Lawrence radiographic grades, MMPs, osteoarthritis, synovial-fluid, synovitis

## Abstract

Failure of conventional anti-inflammatory therapies in osteoarthritis (OA) underlines the insufficient knowledge about inflammatory mechanisms, patterns and their relationship with cartilage degradation. Considering non-linear nature of cartilage loss in OA, a better understanding of inflammatory milieu and MMP status at different stages of OA is required to design early-stage therapies or personalized disease management. For this, an investigation based on a synovium-synovial fluid (SF) axis was planned to study OA associated changes in synovium and SF along the progressive grades of OA. Gene expressions in synovial-biopsies from different grades OA patients (N = 26) revealed a peak of IL-1β, IL-15, PGE2 and NGF in early OA (Kellgren–Lawrence (KL) grade-I and II); the highest MMP levels were found in advanced stages (KL grade-III and IV). MMPs (MMP-1, 13, 2 and 9) abundance and FALGPA activity estimated in forty SFs of progressive grades showed the maximum protein levels and activity in KL grade-II and III. In an SF challenge test, SW982 and THP1 cells were treated with progressive grade SFs to study the dynamics of MMPs modulation in inflammatory microenvironment; the test yielded a result pattern, which matched with FALGPA and the protein-levels estimation. Inflammatory mediators in SFs served as steering factor for MMP up-regulation. A correlation-matrix of IL-1β and MMPs revealed expressional negative correlation.

## 1. Introduction

Osteoarthritis (OA) is the most prevalent whole-joint disorder that exists in the elderly population. Although, OA pathology is not fully understood, it is a multifactorial disorder marked by several cellular and molecular changes, such as an imbalance between cartilage anabolism and catabolism, chondrocytes hypertrophy and death, infiltration of macrophages and activation of immune responses and synovial inflammation and hypertrophy. All together, these pathological changes lead to a gradual loss of articular cartilage. Clinical symptoms, such as joint stiffness, warmth, pain, and joint effusion indicate a presence of inflammation in OA joints [1]. Evolving comprehension of the pathology revealed that OA inflammation is particularly of a low-grade and fundamentally differs from inflammation in rheumatoid arthritis (RA). Scientific studies reported that pro-inflammatory markers in synovial fluid (SF) and blood from OA patients were modest in comparison with RA individuals but higher than healthy controls [2]. Arthroscopy and histological investigation of OA synovium revealed a sub-acute type of inflammation, which is less pronounced than in RA synovium [3]. Further, immune cells infiltration (like macrophages and T cells) in OA synovium is much lower, when compared to RA synovium [4]. The gathered evidence suggests that mononuclear infiltration and over expression of inflammatory mediators in synovium are seen in early OA and predate radiographic damage in OA [5]. Immunopathological studies in OA revealed that inflammation is primarily responsible for triggering innate immune responses that further lead to activation of metalloproteinases (MMPs), in part [4]. Predisposition of obese and diabetic individuals to OA also suggests that systemic inflammatory factors may play an important role in the pathology [6]. Thus, a strong involvement of inflammatory component is increasingly recognized in OA. 

Considering an evolving role of inflammation in OA, a systemic blocking of conventional inflammatory pathways (such as anti-TNF or anti-IL1β agents) was thought to be effective; however, it was found that there was no benefit in generalized OA and minimally effective in erosive OA [7,8]. This underlines a fact that inflammatory mechanism in OA is different and not fully known. Allied to this, further illumination is also needed on the causal relationship between inflammation and cartilage degradation process to make anti-inflammatory therapies more effective in OA management. Synovial membrane and synovial fluid (SF) are the major contributors of inflammation. Key cytokines including interleukin-1β (IL-1β), tumor necrosis factor-α (TNF-α), IL-15, IL-6, IL-17 are produced by synoviocytes and macrophages, the membrane cells. Most of these cytokines including IL-1β, TNF-α and IL-15 are the chief regulators of MMPs, which are primarily responsible for cartilage loss in OA [9]. Moreover, synovium is known to translate systemic inflammation into OA specific pathology. On the other hand, SF is a biochemical channel that transmit and receive molecular signal in the joint cavity. Therefore, focusing on synovium-SF axis can unravel inflammatory network in OA. Recent research studies showed that cartilage loss is reversible, if intervened at the early stages of the disease [10]. Accounting non-linear progressive cartilage loss and chronic nature of OA a need is imposed to understand inflammatory milieu and MMP status at different stages of OA for the development of early-stage therapies or personalized disease management. 

In this communication, we present a grade-wise pattern of key inflammatory agents and MMPs, estimated in synovium, collected from different grades of OA patients. Further, OA specific MMPs (MMP-1, MMP-13, MMP-2 and MMP-9) levels were assessed in OA SFs to find out their activity and abundance at various stages during the disease progression. To understand effect of inflammatory microenvironment on MMPs modulation, SF challenge test was performed. In this test, OA SFs of different grades were used to induce inflammation in human synoviocytes and monocytes; after the SF treatment MMPs modulation was recorded. Finally, a correlation matrix of cytokines and MMPs was developed to find out functional association between inflammation and cartilage degeneration in OA. In all, the outcomes of this multilevel analysis of inflammatory factors and MMPs in OA clinical samples are expected to enlighten the existing knowledge about the perpetuating inflammatory mediators and their effects on MMPs. Improved understanding will ultimately boost the efforts to develop the disease stage-wise or personalized management for OA.

## 2. Materials and Methods

### 2.1. Ethical Statement 

The methods of this research work abide by the Declaration of Helsinki and the protocols were approved by the Institutional Ethics Committee (BVDU/MC/01; Date of approval—15 June 2015). All the study patients signed an informed written consent before participation.

### 2.2. Collection of Clinical Samples

#### 2.2.1. Synovial Biopsies

Synovial biopsies (*n* = 26; KL grade-I = 6; KL grade-II = 6; KL grade-III = 6; KL grade-IV = 8) were collected from OA patients, who had undergone knee arthroscopy or knee replacement surgery. OA diagnosis was performed by experienced rheumatologist based on clinical symptoms and radiographs. OA grading was determined by commonly used radiographic Kellgren–Lawrence (KL) score. According to this score system, grade-I—doubtful OA with presence of minor osteophytes of doubtful importance, grade-II—minimal OA, with definite osteophytes but unimpaired joint space, grade-III—moderate OA, with osteophytes and moderate diminution of joint space whereas grade -IV—severe OA, with greatly impaired joint space and sclerosis of subchondral bone [11]. 

The collected biopsies were stored in liquid nitrogen before they were used for inflammatory biomarkers and MMPs expression analysis.

#### 2.2.2. SF Sample Collection

SF samples (*n* = 40; KL grade-I = 9; KL grade-II = 14; KL grade-III = 10, KL grade-IV = 7) were collected from different grades of OA patients by knee arthrocentesis as described [12]. The fluid collection process was performed in minor operation theater under strict aseptic conditions to collect SFs from early grade OA patients. The fluids from advanced grade OA were collected at the time of knee replacement surgery. The collected samples were stored in −80 °C and were diluted in 1X PBS in the ratio 1:10 before they were used for the planned assays.

#### 2.2.3. Estimation of Pro-Inflammatory Markers and MMPs in the Synovial Biopsies

IL-1β, IL-15, prostaglandin-E2 (PGE2), nerve growth factor (NGF), MMP-1, MMP-13, and TIMP-1 expression pattern was determined in the collected synovial biopsies. First, RNA isolation was performed in the collected tissues, using PureLink RNA mini kit (Invitrogen, CA, USA) as per the manufacturer’s instructions. The tissues were homogenized in 1 mL Trizol reagent for 5 min. Later, 200 µL chloroform was added in the same tube, vortexed for 15 s and kept for 3 min at RT. This step was followed by centrifugation at 12,000 rpm for 15 min at 4 °C. 400 µL supernatant was transferred in a fresh tube and equal volume of ethanol was added. This mixture was later transferred in batches to a spin cartridge and was centrifuged for 30 s at RT. Flow through was discarded and this step was repeated until the whole sample got processed. The membrane bound RNA was then subjected to thorough washing; 700 µL of Wash Buffer-I was added to spin cartridge and was centrifuged at 12,000× *g* at RT for 30 s. The spin cartridge was then shifted to another collection tube and washed twice by adding 500 µL of wash buffer, centrifuged at 12,000× *g* at RT for 30 s and the flow through was discarded. Further, the spin cartridge was centrifuged at 12,000× *g* at RT for 1 min to dry the membrane bound RNA; the collection tube was discarded. For the elution of the bound RNA, the spin cartridge was transferred to a fresh recovery tube and 30 µL RNAase free water was added. After a brief incubation for 2 min at RT, the spin cartridge was centrifuged at 12,000× *g* at RT for one minute. Quality of eluted RNA was determined by using denaturing agarose gel electrophoresis and the quantification was done by measuring absorbance at 260 nm. This RNA was later used for cDNA synthesis, as explained later in this section.

#### 2.2.4. IL-1β Estimation in SF Samples

IL-1β estimation in the collected OA SFs were performed using commercially available ELISA kit (Abnova, Walnut, CA, USA) and following the manufacturer’s instructions. The SF samples included for IL-1β levels determination were thirty-three (KL grade-I = 8; KL grade-II = 7; KL grade-III = 6; KL grade-IV = 12). The optical density was read at 450 nm using ELISA microplate reader (Biorad, Hercules, CA, USA). 

#### 2.2.5. MMP Levels Estimation in SF Samples

Abundance of MMP-1, MMP-2, MMP-9, MMP-13 and TIMP-1 in their protein form was estimated in the collected SFs using commercially available ELISA kits (Cloud-Clone Corp, Houston, TX, USA) and as per the manufacturer’s instructions. In brief, 100 µL of standard diluent was added in antibody coated well as a blank and prepared 7 concentrations for standards; then diluted SF samples were added in antibody coated wells and incubated for 1 h at RT. In case of TIMP-1, the sample incubation period was 2 h at RT. After the incubation, the samples were removed from wells and 100 µL of detection reagent-A was added in each well. The plate was covered with a plate sealer and again incubated for 1 h at 37 °C. After incubation, each well was washed thrice with 350 µL wash buffer. After the washing procedure, 100 µL of detection reagent-B was added in each well and incubated for 30 min at 37 °C. On completion of incubation, the washing procedure was performed 5 times using washing buffer. Later, 90 µL of substrate solution was added in to each well and was incubated in dark for 10–20 min at RT. Finally, 50 µL stop solution was added and read at 450 nm immediately. MMP and TIMP levels were calculated by using standard graph and multiplied by dilution factor to get the final value.

### 2.3. Collagenase Activity of MMPs, Estimated in SF Samples Using FALGPA Reagent

Collagenase activity of MMPs was determined using FALGPA, a synthetic substrate, which has resemblance to the primary structure of collagen and is hydrolyzed by all known collagenases. For FALGPA assay, 200 µL of FALGPA reagent was added into a mixture of 10 µL SF sample and 90 µL of Tris buffer. This mixture was further incubated for 15 min at 25 °C. Post-incubation, the absorbance was measured at 340 nm using Perkin Elmer UV/VIS spectrophotometer. The same procedure was repeated in another tube with 100 µL of buffered collagenase enzyme, which was used as a control. FALGPA unit is hydrolysis of 1.0 micromole of FALGPA per min at 25 °C pH 8.8 in a presence of calcium ions. 

### 2.4. SF Challenge Test on Human Synoviocytes (SW982) and Monocytes (THP1)

This experiment was planned to assess MMP specific response generated by inflamed SW982 and THP1 cells. Both the cell lines were obtained from National Centre for Cell Science (NCCS), Pune. SW982 cells were grown in culture media composed of dulbecco’s modified eagle medium (DMEM) + 10% fetal bovine serum (FBS) + 2 mmol/L L-glutamine + 100 U/mL penicillin + 100 µg/mL streptomycin. On the other hand, THP1 cells were maintained in Roswell Park Memorial Institute (RPMI) 1640 medium (containing10% FBS + 100 U/mL penicillin + 100 µg/mL streptomycin). Both the cells were maintained and grown at 95% relative humidity and 5% CO_2_ at 37 °C. 

During SF challenge test, SW982 were treated with different grades of OA SFs (5% of culture media) for 72 h, while THP1 cells were treated with 10% SF (of culture media) for 48 h to induce inflammation. Successful induction of inflammation was confirmed by estimation of nitric oxide (NO) and IL-1β. NO release was estimated from the cell culture medium, using Griess reaction method as described [13]. IL-1β expression in the SF treated SW982 was tested by qPCR and the values were normalized against β-actin gene. SF challenge test was developed and standardized by our research group and was successfully used to study OA specific responses [14]. 

### 2.5. cDNA Synthesis

After the SF treatment, both the cells were harvested for RNA isolation. RNA isolation was done using PureLink RNA mini kit (Invitrogen, Carlsbad, CA, USA) as per the manufacturer’s instructions. The cells without SF treatment were used as a control. Isolated RNA was used for cDNA synthesis, which was done by using a high-capacity cDNA reverse transcription kit (Invitrogen, CA, USA). As per the manufacturer’s instructions, for each 20 µL reaction, 2 µg of isolated RNA was used. Depending on the number of reactions, the reagents’ volumes were calculated. For preparation of one reaction of 2X RT master mix, 2.0 µL of 10X RT Buffer, 2.0 µL of 10X RT Random primers, 0.8 µL of 25X dNTP mix (100 mM), 1 µL of MultiScribe™ Reverse Transcriptase and 4.2 µL of nuclease-free water was used. The reaction mixture was kept on ice and mixed gently. In total, 10 µL of RNA was added into 2X RT master-mix to prepare cDNA RT reaction in a PCR tube and was mixed gently by pipetting up and down. Finally, a PCR tube was centrifuged to spin down all the contents and to remove air bubbles if any. The tube was then placed in a thermal cycler that was programmed according to manufacturer’s instructions. The reverse transcriptase run cycle was performed as described. On the completion of one cycle, the obtained cDNA was stored at −80 °C for quantitative polymerase chain reaction (qPCR) analysis.

### 2.6. qPCR Analysis

qPCR analysis was performed using Applied Biosystems Step One Real Time PCR System as per the manufacturer’s instructions (Applied Biosystems, Foster City, CA, USA). TaqMan gene expression assays (Applied Biosystems, Foster City, CA, USA) analyzed in the SF treated SW982 and THP1 cells were, MMP1, MMP-13. Additionally, VEGF-1 expression was determined in the SF treated THP1 cells. mRNA levels of the selected genes were calculated to the amount of β-actin using Step One Software version 2.2.2.

All the reagents, chemicals and cell/tissue culture media used in the experimental work were of analytical grade and were purchased from Sigma-Aldrich (St. Louis, MO, USA). Plastic-ware was procured from BD Biosciences and Axygen Scientific Inc., Union City, CA, USA. 

### 2.7.Statistical Analysis

The clinical sample evaluation for the inflammatory factors and MMP estimation and in vitro studies were performed in triplicates and the data was presented as mean ± SD. Inter-grade statistical significance was calculated by One Way ANOVA followed by a Kruskal–Wallis test and Tukey’s test of significance using R software version 3.3.0.

## 3. Results

### 3.1. Synovial Biopsies Gene Expression Study 

The highest mRNA level of IL-1β and IL-15 expression was found in KL grade-I synovium samples. Thereafter, a drop was noticed in the expression levels of both the cytokines with a trivial increase in KL grade-IV biopsies. During inter-grade comparison a significant difference in IL-1β expression was noted between KL grade-I and II (*p* = 7.70 × 10^−3^; *p* < 0.05); the difference among all the other KL grades was found insignificant. IL-15 expression showed a significant difference for KL grade-I vs. III (*p* = 3.66 × 10^−2^; *p* < 0.05) (Figure 1 A,B). We did not find any particular expression trend for PGE2 and NGF expression and maximum expression level of PGE2 and NGF was found in KL grade-II and grade-IV biopsies respectively. Both the inflammatory agents did not reveal any significant observation during inter-grade comparison (Figure 1C,D).

MMP-1 showed a grade-wise decline in the expression; the highest MMP-1 expression was found in early OA samples (KL grade-I). The expression levels in KL grade-III and IV Synovial samples were comparable (Figure 1E). In contrast to MMP-1, MMP-13 revealed a grade-wise elevation in the expression level. Peak expression was found in KL grade-III biopsies and the least expression levels was detected in KL grade-I (Figure 1F). TIMP-1 expression was comparable in all the synovial biopsies (Figure 1G). Both the MMPs and TIMP-1 did not show any significant difference in the expression level for inter-grade comparison.

### 3.2. OA SFs Analysis 

#### 3.2.1. Estimation of IL-1β Levels in SFs

IL-1β protein levels were determined in different degree of OA SFs and are presented in Figure 2A. The highest IL-1β was estimated in KL grade-I with the average 202.16 pg/mL, while the least was noted in grade-III sample with the average 36.77 pg/mL. The mean IL-1β for KL grade-II and IV was 56.45 pg/mL and 48.37 pg/mL, respectively. A grade-wise comparison of IL-1β revealed a difference in the level as—grade-I vs. grade-II (*p* = 0.20295), grade-I vs. grade-III (*p* = 0.14694), grade-I vs. grade-IV (*p* = 0.09531), grade-II vs. grade-III (*p* = 0.99411), grade-II vs. grade-IV (*p* = 0.99933) and grade-III vs. grade-IV (*p* = 0.99831). Thus, no statistically significant difference was noted among the KL grades.

#### 3.2.2. Estimation of MMP Abundance and Activity in SFs

MMP-1, MMP-13, MMP-2 and MMP-9 protein abundance was estimated in OA SFs. The highest MMP-1 was seen in KL grade-II (average—35.22 ng/mL) and grade-III SF samples (average—34.58 ng/mL), while the least was observed in KL grade-I samples (average—22.90 ng/mL). In the grade-wise comparison, significant differences were noted between KL grade-I and II (*p* = 6.2 × 10^−4^; *p* < 0.001), KL grade-I and III (*p* = 2.73 × 10^−3^; *p* < 0.01) and KL grade-II and IV (*p* = 2.71 × 10^−2^; *p* < 0.05). The difference between KL grade-III and IV (*p* = 6.49 × 10^−2^), KL grade-I and IV (*p* = 0.77028) as well as KL grade-II and grade-III (*p* = 0.99547) remained statistically insignificant (Figure 2B).

For MMP-13, the maximum increase was seen with KL grades-II (average—5.85 pg/mL) and grade-III SF samples (average—5.45 pg/mL). KL grades-I (average—2.53 pg/mL) and grade-IV samples (average—2.24 pg/mL) interestingly showed comparable MMP-13 levels. A noteworthy difference was found between KL grades-I and II (*p* = 6.32 × 10^−5^; *p* < 0.001), KL grades-I and III (*p* = 9.72 × 10^−4^, *p* < 0.001) and between KL grades-IV and II was outstanding and was significant at 1% level (*p* = 5.13 × 10^−4^; *p* < 0.001). A difference between KL grades-IV and III was also significant (*p* = 4.86 × 10^−3^; *p* < 0.01). However, the difference between KL grade-I and IV (*p* = 0.9921594) and between KL grade-II and III (*p* = 0.9247271) was insignificant (Figure 2C). 

MMP-2 level in OA SF samples followed a trend like MMP-1 and MMP-13; the highest MMP-2 was found in KL grade-III samples (average—8.92 pg/mL). The least MMP-2 was noted in KL grade-I SFs (average—2.70 pg/mL). A noteworthy difference in MMP-2 was observed between KL grade-I and II (*p* = 1.71 × 10^−3^; *p* < 0.001), KL grade-I and III (*p* = 3.61 × 10^−8^; *p* < 0.001) and between KL grade-I and IV (*p* = 1.86 × 10^−5^, *p* < 0.001). An outstanding difference was found in between KL grades-II and III (*p* = 7.40 × 10^−4^; *p* < 0.001); however, the difference between KL grades-II and IV (*p* = 0.1148841) and KL grade-III and IV (*p* = 0.4928022) was not significant (Figure 2D). 

MMP-9 showed the highest level in moderate OA SFs (KL grade-II: average—6.14 ng/mL, KL grade-III: average—6.80 ng/mL). During inter-grade comparison, a significant elevation was noted in KL grades-II (*p* = 0.02494) and III (*p* = 0.01368), when compared with KL grade-I (*p*< 0.05). However, variation among all the other grades (KL grade-I vs. grade-IV (*p* = 0.15460); KL grade-II vs. grade-III (*p* = 0.95931); KL grade-II vs. grade-IV (*p* = 0.97699); KL grade-III vs. grade-IV (*p* = 0.85443)) was insignificant (Figure 2E). 

TIMP-1 level was high in KL grades-I (average—6.81 ng/mL) and II SF samples (average—6.72 ng/mL), while it was dropped significantly in KL grade-III (average—3.46 ng/mL) and IV (average—3.52 ng/mL). During inter-grade comparison, a marked difference was found between KL grades-I and III (*p* = 3.30 × 10^−4^; *p* < 0.001) as well as between KL grades-I and IV (*p* = 1.35 × 10^−3^; *p* < 0.001). Similarly, KL grades-II and III (*p* = 1.10 × 10^−4^; *p* < 0.001) as well as KL grades-II and IV (*p* = 6.70 × 10^−4^; *p* < 0.001) showed an outstanding difference in the TIMP-1. However, the difference between grade-I and II (*p* = 0.99917) and grade-III and IV (*p* = 0.99978) was insignificant (Figure 2F).

#### 3.2.3. Collagenase Activity of MMPs Estimated in OA SFs Using FALGPA Substrate

The results of this assay were presented in the form of collagenase activity units achieved by synthetic substrate of FALGPA (Figure 2G). The maximum value for MMPs activity was noted in moderate grade OA SFs (KL grades-II and III) and the lowest was observed in early OA samples (KL grade I). The highest average FALGPA was estimated in KL grade IV samples (average—34.85 collagenase activity units). During inter-grade comparison, the difference revealed in collagenase activity was KL grade-I vs. grade-II (*p* = 0.99873), KL grade-I vs. grade-III (*p* = 0.62688), KL grade-I vs. grade-IV (*p* = 0.29002), KL grade-II vs. grade-III (*p* = 0.64351), KL grade-II vs. grade-IV (*p* = 0.27758) and KL grade-III vs. grade-IV (*p* = 0.92185); no statistical difference was noted among the grades. 

### 3.3. SF Challenge Test 

SW982 cells

Inflammation induction test

NO release estimation 

A grade-wise increase in NO levels was observed after 72 h of SF induction. Maximum inflammation was found in the cells treated with KL grade-III SFs, with a slight decline after the treatment with grade IV-SF samples. A significant difference in NO estimation, observed during inter-grade comparison, was denoted in Figure 3A. 

### 3.4. IL-β Expression

Elevation in IL-β expression was noticed in the cells after 72 h of SF treatment. Treatment of KL grade-I and II showed a marginal up-regulation of IL-1β; the highest rise in the expression was seen, when the cells were treated with KL grade-III SFs. During inter-grade comparison, IL-1β revealed differences as—grade-I vs. grade-II (*p* = 0.917), grade-I vs. grade-III (*p* = 0.0520), grade-I vs. grade-IV (*p* = 0.975), grade-II vs. grade-III (*p* = 11.01 × 10^−2^; *p* < 0.05), grade-II vs. grade- IV (*p* = 0.713) and grade-III vs. grade-IV (*p* = 0.130) (Figure 3B).

### 3.5. MMP Specific Response of the Inflamed SW982

MMP specific response was estimated in the form of mRNA levels of MMP-1 and MMP-13. SF induced SW982 cells revealed a grade-wise increase in MMP-1 expression with a decline in the expression level in the cells treated with grade IV SFs. MMP-1 up-regulation was significant between KL grade I and III (*p* = 6.10 × 10^−8^; *p* < 0.001) as well as KL grade I and IV (*p* = 1.04e^−5^; *p* < 0.001). A significant difference in MMP-1 was also noted between KL grade II and III (*p* = 2.63 × 10^−6^; *p*< 0.001) and KL grade II and grade IV (*p* = 8.48 × 10^−4^; *p* < 0.001). The difference between KL grade-III and IV (*p* = 0.0658), as well as KL grade-I and II (*p* = 0.223), was recorded insignificant (Figure 3C). 

MMP-13 showed a similar pattern of MMP-1 and revealed a grade-wise increase in the expression; the highest expression was noted in the cells challenged with KL grade IV SFs. The difference in the expression level was notably significant between KL grade I and II (*p* = 9.16 × 10^−8^; *p* < 0.001) and KL grades I and III (*p* = 8.12 × 10^−7^; *p* < 0.001) and KL grade I and IV (*p* = 6.30 × 10^−11^; *p* < 0.001). A significant difference was also found between KL grade-II and IV (*p* = 6.56 × 10^−4^; *p* < 0.001) and KL grade-III and IV (*p* = 4.77 × 10^−5^; *p* < 0.001) (Figure 3D). There was no difference in the expression level for MMP-13 in between KL grade-II and III (*p* = 0.651).

### 3.6. THP1 Cells

Inflammation induction test

NO release estimation

NO estimation in SF treated THP1cells showed a similar pattern to SW982 cells; a grad wise increase was noted in the NO release. The highest NO release was measured in the cells treated with KL grade-II and II grade SFs, when compared to control cells. A grade wise difference in NO estimation and statistical significance is showed in Figure 4A.

### 3.7. MMP Specific Response of the Inflamed THP1

Like SW982, MMP specific response of the inflamed THP1 was estimated in the form of mRNA levels of MMP-1 and MMP-13. Additionally, we assessed mRNA levels of VEGF-1 on these cells.

The highest MMP-1 expression was found in the cells treated with SF of KL grade II and was followed by a grade-wise decline thereafter. In inter-grade comparison, the cells treated with KL grade-I SF showed a significant difference in MMP-1 expression, when compared to KL grade-II (*p* = 1.93 × 10^−14^; *p* < 0.001) and KL grade-III (*p* = 5.48 × 10^−6^; *p* < 0.001). The difference in the expression was also noted between the cells treated with SF of KL grade-II and III (*p* = 2.80 × 10^−14^; *p* < 0.001), KL grade-II and IV (*p* = 1.92 × 10^−14^; *p* < 0.001) and KL grade-III and IV (*p* = 1.37 × 10^−6^; *p* < 0.001). Comparison of MMP-1 expression between KL grade-I and IV was marked as insignificant (*p* = 0.901) (Figure 4B). 

MMP-13 expression in THP1 revealed a grade-wise increase. Maximum expression was found in the cells, which were treated with KL grade IV. Except the comparison between SF treatment with KL grade-II and grade-III (*p* = 0.923), a significant difference in MMP-13 expression was found in grade-wise comparison (KL grade-I vs. II (*p* = 3.91 × 10^−2^; *p* < 0.05), KL grade-I vs. III (*p* = 1.01 × 10^−2^; *p* < 0.05), KL grade-I vs. IV (*p* = 1.19 × 10^−12^; *p* < 0.001), KL grade-II vs. IV (*p* = 3.76 × 10^−11^; *p* < 0.001), KL grade-III vs. IV (*p* = 8.45 × 10^−11^; *p* < 0.001)) (Figure 4C). 

The highest expression of VEGF−1 was noted in the cells treated with SFs of KL grade II; this was followed by a grade-wise decline in the expression. Therefore, VEGF−1 expression pattern was similar to MMP-1 expression. Inter-grade comparison for VEGF−1 revealed a marked difference in KL grade-I vs. II (*p* = 1.92 × 10^−14^; *p* < 0.001), KL grade-I vs. III (*p* = 2.19 × 10^−14^; *p* < 0.001), KL grade-I vs. IV (*p* = 8.58 × 10^−8^; *p* < 0.001), KL grade-II vs. III (*p* = 8.53 × 10^−6^; *p* < 0.001), KL grade-II vs. IV (*p* = 3.44 × 10^−14^; *p* < 0.001) and KL grade-III vs. IV (*p* = 1.89 × 10^−11^; *p* < 0.001) (Figure 4D).

### 3.8. Functional Co-Relation between IL-1β and MMPs 

To explore a functional relationship between IL-1β, a key OA cytokine and MMPs, a correlation matrix was plotted using a cumulative mean value (mean of all grades) estimated from SF for each selected marker (Figure 5). It revealed a strong negative correlation between IL-β and MMPs. In particular, the cytokine showed a strong negative correlation with MMP-9 (correlation coefficient: −0.98) and MMP-2 (correlation coefficient: −0.92). A moderate negative correlation was found between IL-1β and MMP-1 (correlation coefficient: −0.74) and between IL-1β and MMP-13 (correlation coefficient: −0.52). The matrix also showed a strong positive correlation between MMP-1 and MMP-13 (correlation coefficient: 0.96) as well as between MMP-2 and MMP-9 (correlation coefficient: 0.90).

## 4. Discussion

The present study investigates various molecular and cellular events along the synovium-synovial fluid axis, as it plays the unique significant biological functions of nurturing, maintaining and protecting cartilage. This axis is also known for its barrier shielding function for cartilage from various biochemical insults and obligating a protective role. All the systemic signals are received by synovium and relayed by SF and hence are expected to hold early to advance signs of the disease pathology [9]. Gene expressions in synovium biopsies in this work formed a purposive approach to study the variation pattern of inflammatory mediators and MMPs during natural OA progression. A definite expression trend was noticed in the selected markers. Peak expressions of IL-1β and IL-15 in KL grade-I synovium indicated that the regulators of the inflammation were expressed higher at early stage that provides a biochemical trigger even before appearance of undoubted clinical signatures of OA. The highest PGE2 was found in KL grade-II samples, immediately in the next stage of peak IL-1β, the main inducer of PGE2. NGF showed comparable expressions in KL grades-I, III and IV. In a grade-wise comparison, IL-1β and IL-15 expression showed a marked significance in KL grade-I synovium. However, the other inflammatory markers and MMPs did not show any significant difference, possibly due to a smaller number of samples. Taken together, the expression trend of these pro-inflammatory markers explained persistent synovitis in all stages of OA, unlike a common claim that synovial inflammation is prevalent at early and advanced stages. Further, these results indicated that each stage of OA was marked by a presence of a particular inflammatory factor. The individual role of these mediators has been extensively studied in OA and their increased levels has been recorded in OA affected clinical samples [15,16,17,18]. Combined effect of elevated IL-β, IL-15 and PGE2 in early stages of OA provided the necessary impetus for accelerated cartilage loss at later stages of the disease as evident by relatively higher expression of MMPs in advanced staged synovial biopsies (KL grade-III and IV). Of note, excessive levels of IL-β, IL-15 and PGE2 are associated with impaired bone and cartilage remodeling. Particularly IL-1β and IL-15 are known to cause a destructive effect on articular cartilage by up-regulating MMPs [16,19]. PGE2 and NGF, both are pro-angiogenic factors and their high levels contribute to synovial inflammation and hyperplasia. Additionally, NGF has a strong connection with transmission of pain in OA [18,20]. On this background, comparable levels of TIMP−1 in all grade synovium membranes can be a natural countering against increasing MMP activity to limit the cartilage-loss. Besides, the data presented here give a context of the progressive grades of OA, which has not been adequately represented in the published literature so far.

FALGPA and ELISA assays represented the collagenolytic MMPs activity and their protein abundance in SF, respectively. All MMP protein levels were noted higher in moderate grade SFs (KL grade-II and III) whereas, their collagenolytic activity as measured through FALGPA assay was found steadily increased in SF with advanced grades. TIMP is a natural inhibitor secreted by tissues and its abundance is considered to be in repulsion with the concentration of MMPs. In our observation here, TIMP abundance clearly decreased in SF of KL grade-III and IV with concomitant increase in MMPs (Figure 2F,G). In this study a dichotomy was detected in abundance of TIMP−1 in SF and its expression in synovium. Here, a significant reduction of TIMP−1 was noted in late OA SFs (KL grade-III and IV), whereas its expression was consistent in the synovium from all the grades. Although, several explanations for this finding are possible, it is likely that TIMP−1 concentration in SF is not contributed only by the synovium. This may be recorded as limitation of present data that other tissues and other TIMPs were not studied, which certainly contribute to ECM turnover [21]. Furthermore, TIMPs are not the only inhibitors for MMPs; Tchetverikov et al. (2005) showed α2-macroglobulin played a leading role in scavenging activated MMPs. However, the increased proMMP/TIMP ratio in favor of MMPs in their study, correlate with our observation. It may be interesting to note the similarities among progressive OA, knee injury and inflammatory arthritis [22]. A meta-analysis covering a total of 1408 studies designed to study relationship between expression of MMPs and pathogenesis of OA, revealed higher expression of MMP-1, MMP-2 and MMP-9 in OA, as well as ethnic differences in their expression except for MMP-9 [23]. Further, FALGPA and ELISA assay outcomes were also corresponded with our previous results of glycosaminoglycan (GAG) estimation in OA SFs of different grades. Considering GAG as a direct product of MMP action on cartilage, the highest GAG in KL grade-II and III SFs was a clear indication of the maximum activity of MMPs at these grades [24]. Sachdeva et al. [25] reported similar observation on the ethnically similar population as in the present study, wherein MMP-9 and MMP-13 were studied in SF of OA patients. The expression of both the MMPs was reported high in KL grade-II and KL grade-III SFs and subsides in KL grade-IV.

Higher levels of pro-inflammatory factors in SF have been reported by many workers, including some proteomic study wherein proteins and peptides were detected, which may act as trigger for cellular and tissue inflammation. In a study on 100 subjects, Saetan et al. (2014) reported higher VEGF expression in synovial tissue and corresponding increased levels in SF and serum [26]. However, these SF proteins need to be tested for their potential to induce biologically relevant inflammation in joint tissue individually as well as collectively to elucidate the underlining pathological inflammatory mechanism. The novelty of the present data is the SF challenge test that was designed to study the dynamics of modulation in inflammatory microenvironment at joint in terms of classical inflammatory mediators and MMPs expression. These experiments were based on our previous learning that OA SF holds inflammatory milieu of various cytokines and chemokines and can be used to study the disease specific responses on cell lines [14]. Most of the etiopathology of OA spins around the synovium-SF-cartilage axis, wherein SF occupies the central position. The SF challenge test was carried out with the SF obtained from OA patients of different grades and with two different types of cells. The advantage of this biological assay was its physiological relevance; the cells are subjected to their natural inflammatory micro-environment as in OA joints and hence are expected to mimic their response as in vivo conditions. The SF challenge test was performed using human monocytes and synoviocytes, the two major cell types of synovium, which are involved in the synovial inflammation. Successful induction of inflammation in the SF treated cells was confirmed by estimation of NO and mRNA levels of IL-1β (Figure 3A,B and Figure 4A). NO is a known inflammatory marker produced by the cells under stress. In the present work, SF treatment in both the cell lines showed a significant NO release as compared to untreated cells (Figure 3, Table a1 and Figure 4a1). In inflamed joints, this unmitigated NO stimulates excessive MMP production and inhibits synthesis of collagen and proteoglycans [27]. Interestingly, both the cell lines showed up-regulation of MMP-1 and MMP-13 in a similar manner; this means that moderate grade OA SFs (KL grade-II and III) caused the highest up-regulation of MMP-1, while KL grade-IV SF treatment was responsible for the maximum up-regulation of MMP-13. Additionally, SF treatment on THP1 also up-regulated VEGF−1, a pro-angiogenic factor. A surprising data published by Hoff et al. (2013) showed that OA-SF is more potent over to RA-SF in inducing VEGF in primary chondrocytes indicating that SF contains pro-inflammatory mediators capable of inducing inflammatory changes in synoviocytes on one hand and chondrocytes on the other [28]. 

To find out a functional correlation between inflammation and cartilage-loss in OA, we plotted a correlation matrix of IL-1β and MMPs using their values in SF (Figure 5). Of note, SF represents a “collective secretome” of all the cells in the joint that include, synovium, cartilage, meniscus, and immune cells and thus, represents a true cross-section of metabolic status at the given KL grade of the joint. Moreover, in this study, SF values yielded a consistent pattern of correlation matrix, which matched with FALGPA and SF challenge test. This correlation matrix revealed a negative correlation between IL-1β and MMPs, particularly, with MMP-1, MMP-2 and MMP-9, suggesting a clear special difference in their expression. Otherwise, IL-1β is known to induce MMPs secretion through multiple pathways for example, MMP-1 and MMP-13 are stimulated via MAPK/ERK, NF-κB and Wnt−5A signaling [29], whereas up-regulation of Wnt−5A additionally induce MMP-9 as shown in rabbit chondrocytes [30]. In the recent work, IL-1β was also shown to induce MMPs via up-regulation of Notch1 and NICD in chondrocytes [31]. The studies also revealed that effects of IL-1β on synoviocytes and chondrocytes are similar in terms of strong induction of MMP-2 and MMP-13 via activation of RUNX−2 and Wnt/β-catenin as well as MAPK and NF-κB pathways [32,33]. This research evidence clearly exhibited a positive biological correlation between IL-1β and MMPs although, it differentiated in time as revealed by the present data. IL-1β along with other cytokines is also known to be involved in activation, differentiation and proliferation of effector cells, which ultimately produce MMPs [34]. Overwhelming evidence from various experiments elsewhere and in the present study suggest that cartilage-loss in OA is inflammation driven. We also observed a strong positive correlation between MMP-1 and MMP-13 as well as between MMP-2 and MMP-9 (Figure 5). This was quite expected as both the MMPs belong to the collagenase sub-class of the family and are about 50%−55% identical in sequence. Of note, type-II collagen is a poor substrate for MMP-1, while MMP-13 is five to ten times more potent against it. However, different MMPs overtake each other’s role with comparable efficiency to degrade type-II collagen, is the main reason why MMPs inhibition is not a successful therapeutic strategy. 

The strength of this study includes multilevel analysis of OA tissues (synovium and SF) of across the grades of the disease and rational validation of the outcomes using physiologically relevant in vitro model. On concluding remarks, the etiology of OA is complex and multifactorial and involves trauma, age, systemic comorbidities, and heredity. Typical to the complex non-communicable diseases, rigorous analysis of the structure and the complex dynamics of molecular networks across the various tissues, remains as a prerequisite to give a thorough understanding of the disease, which in the case of OA is still a major impediment in the discovery of potential druggable targets, and to design effective therapy. Further, there is a large person-to-person variation as seen elsewhere and in present analysis, indicating perhaps “one for all” medicine may not be a yielding strategy in OA. A personalized approach built through the combination of inhibition of degrading enzymes, reducing inflammation, immuno-modulation, limiting angiogenesis and synovitis and management of pain appears to be the plausible approach in the management of OA.

## Figures and Tables

**Figure 1 genes-12-00989-f001:**
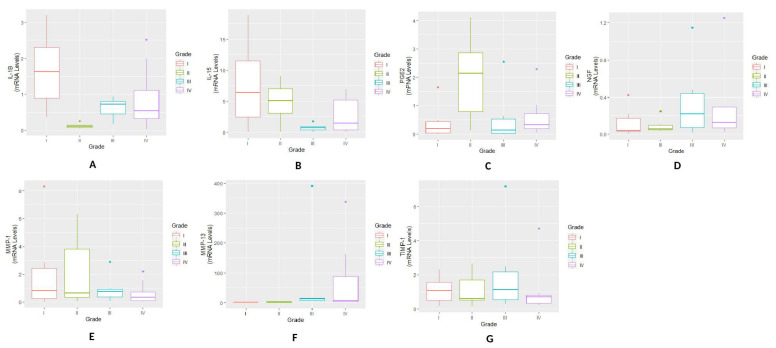
Synovial biopsy gene expression study—various pro-inflammatory markers and MMP expressions were estimated in synovial biopsy samples obtained from different grades of OA patients (*n* = 26) (**A**) a grade-wise expression of IL-1β; the highest expression was found in the samples collected from OA patients with KL grade-I and the lowest expression was noted in KL grade-II samples. A significant difference was noted between KL grade-I and II (*p* < 0.05). (**B**) expression trend of IL-15 showed a grade-wise decline. A trivial increase was noted in KL grade-IV samples; the highest expression was noted in KL grade-I biopsies. A significant difference was noted between KL grade-I and III (*p* < 0.05). (**C**) a grade-wise expression trend of PGE2 with the highest expression in KL grade-II biopsies; inter-grade comparison did not reveal any significant difference among all the KL grades. (**D**) expression pattern of NGF, where the highest expression was seen in KL grade-IV synovial samples; no significant difference was noted among the grades during inter-grade comparison. (**E**) a grade-wise expression trend of MMP-1, where a grade-wise decline was found in the expression; the highest expression of MMP-1 was seen in KL grade-I biopsies. (**F**) expression pattern of MMP-13 showed a grade-wise increase; the highest expression was noted in the KL grade-III. (**G**)TIMP-1 revealed a comparable expression trend in all the KL grades. MMP-1, MMP-13 and TIMP-1 did not reveal any statistically significant difference for inter-grade comparison.

**Figure 2 genes-12-00989-f002:**
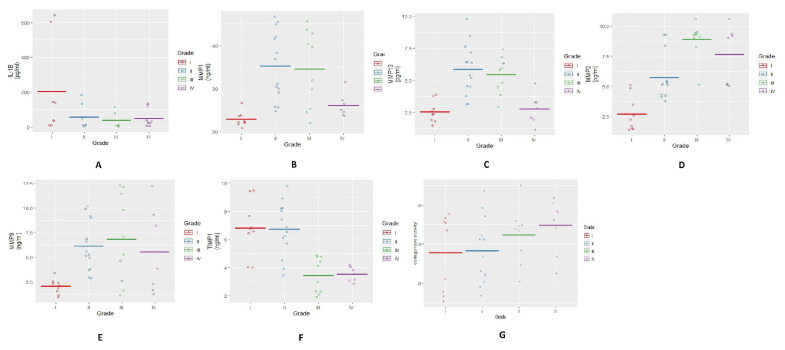
Estimation of MMP abundance (in protein forms) and activity in OA SFs. (**A**) a scatter plot depicting grade-wise estimation of IL-1β; the highest value of IL-1β was found in SFs of KL grade-I. A grade-wise decline was seen in the levels; no statistically significant difference was noted in inter-grade comparison. (**B**) a grade-wise pattern of MMP-1 revealed maximum abundance in SFs from KL grade-II and III; a significant difference in MMP-1 was noted between KL grade-I and II (*p* < 0.001), KL grade-I and grade-III (*p* < 0.01) and KL grade-II and grade-IV (*p* < 0.05). (**C**) depicts grade-wise estimation of MMP-13; the highest levels were found in SFs from KL grade-II and III; statistically significant difference in MMP-13 was noted between KL grade-I and grade-II (*p* < 0.001), KL grade-I and III (*p* < 0.001), KL grade-II and IV (*p* < 0.001) and KL grade-III and grade-IV (*p* < 0.01). (**D**) a grade-wise expression of MMP-2 showed maximum abundance in KL grade-III samples; a marked difference was seen between KL grade-I and II (*p* < 0.001), KL grade-I and III (*p* < 0.001), KL grade-I and IV (*p* < 0.001) and KL grades-II and III (*p* < 0.001). (**E**) a grade-wise estimation of MMP-9 revealed the highest level in moderate OA SFs (KL grade-II and grade-III). In inter-grade comparison, a significant difference was noted between KL grade-I and KL grade-II (*p* < 0.05) and also between KL grade-I and grade-III (*p* < 0.05). (**F**) demonstrates a grade-wise estimation of TIMP-1 level; higher levels were seen in early grade samples (KL grade-I and II); in inter-grade comparison, a marked difference was noted between KL grade-I and III (*p* < 0.001), KL grade-I and IV (*p* < 0.001), KL grade-II and III and KL grade-II and IV (*p* < 0.001). (**G**) shows a grade-wise MMP activity measured using synthetic substrate FALGPA; maximum activity was estimated in SFs from KL grade-II and III, while the lowest activity was found in KL grade-I SFs. No significant difference was noted in inter-grade comparison.

**Figure 3 genes-12-00989-f003:**
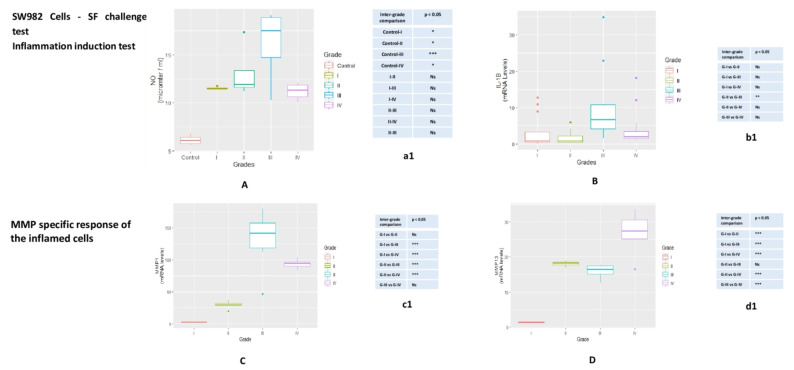
SF challenge test on SW982 cells—this test was performed to assess MMPs modulation in inflamed SW982. The cells were treated with 5% SFs (of culture media) of progressive OA grades for 72 hrs to induce inflammation. Inflammation induction test—(**A**) NO release after 72 hrs of SF treatment on SW982 cells (**B**) mRNA levels of IL-β estimated in SW982 cell after 72 hrs SF treatment. This set of experiments was performed as a confirmation test for successful inflammation induction in the cells after the treatment with SFs from progressive OA patients. For SW982 cells, statistical difference in NO and IL-1β levels observed during inter-grade comparison and compared to the control is showed in tables a1 and b1 respectively. MMP specific response of the inflamed cells—(**C**) MMP-1: a grade-wise up-regulation was noted, where the highest increase in the expression level was found in the cells, which were treated with SFs from KL grade-III; (**D**) MMP-13: a grade-wise increase in the expression was noted with the highest expression in the cells treated with SFs from KL grade-IV. The difference revealed during inter-grade comparison for MMP-1, MMP-13 on SW982 cells is showed in the tables—c1 and d1, respectively. * *p* < 0.05, ** *p* < 0.01, *** *p* < 0.001. Ns = non-significant.

**Figure 4 genes-12-00989-f004:**
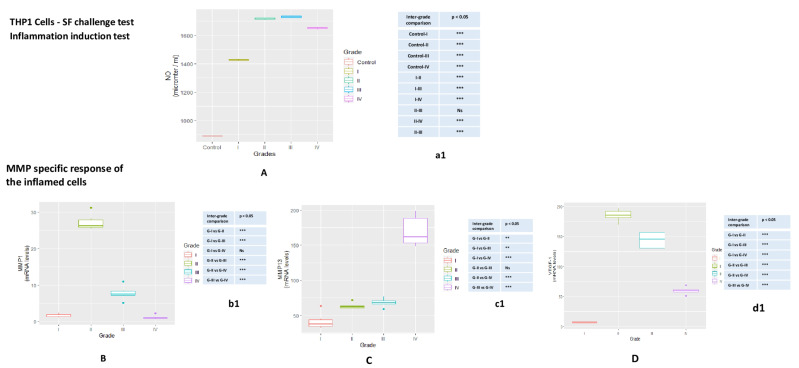
SF challenge test on THP1 cells—the test was carried out to assess MMPs modulation in inflamed THP1. These cells were treated with 10% SFs (of culture media) of progressive OA grades for 48 hrs in order to induce inflammation. Inflammation induction test—(**A**) NO release after 48 hrs of SF treatment on THP1 cells; NO significance, revealed during inter-grade comparison is showed in table a1. MMP-specific response of the inflamed cells—(**B**) MMP-1: the highest rise in the expression in the cells treated with SFs from KL grade-II; (**C**) MMP-13: a grade-wise elevation was seen in MMP-13 expression; the maximum up-regulation was found in the cells treated with SFs from KL grade-IV; (**D**) VEGF: the highest estimation of VEGF was in the cells treated with KL grade-II SFs that was followed by a grade-wise decline in the expression; the lowest expression was seen in the cells treated with SFs from KL grade-I. Inter-grade difference for MMP-1, MMP-13 and VEGF−1 on the cells is showed in tables b1, c1 and d1, respectively. * *p* < 0.05, ** *p* < 0.01, *** *p* < 0.001.

**Figure 5 genes-12-00989-f005:**
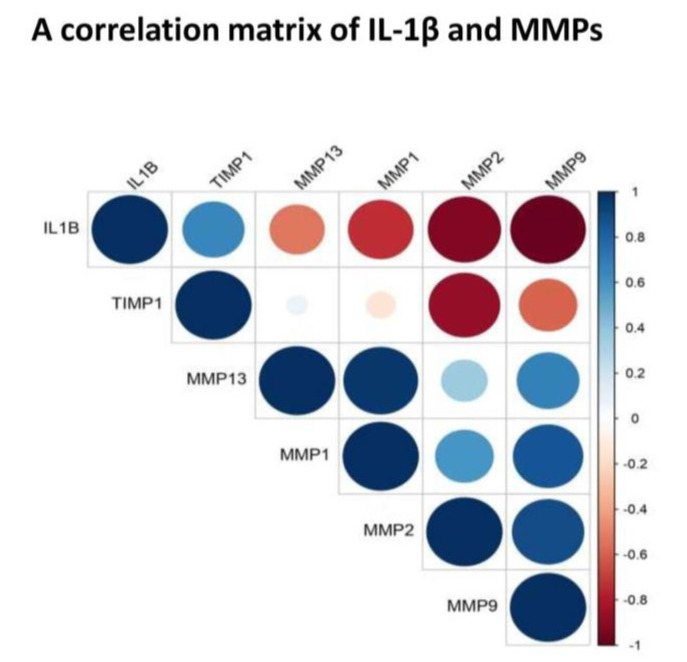
Correlation matrix of IL-1β, MMP-1, MMP-13, MMP-2, MMP-9 and TIMP−1. The matrix was developed using a cumulative mean value (mean of all grades) estimated from SF for each selected marker. It revealed a negative correlation between—IL-β and MMP-1 (correlation coefficient: −0.74); IL-1β and MMP-2 (correlation coefficient: −0.92); IL-1β and MMP-9 (correlation coefficient: −0.98); IL-1β and MMP-13 (correlation coefficient: −0.52); It showed a strong positive correlation between MMP-1 and MMP-13 (correlation coefficient: 0.96) as well as between MMP-2 and MMP-9 (correlation coefficient: 0.90).

## Data Availability

Not applicable.

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
