# Peer review of "Synovium-Synovial Fluid Axis in Osteoarthritis Pathology: A Key Regulator of the Cartilage Degradation Process"

_genes, 2021, doi:10.3390/genes12070989_

Round 1

Reviewer 1 Report

Title: Synovium-synovial fluid axis in osteoarthritis pathology: a key regulator of cartilage degradation process. This manuscript focuses on inflammatory mechanism, pattern and its causal linkage with cartilage degradation. However, this manuscript is lacking in the study in inflammatory mechanism, but with substantial revision could be made suitable for publication.

  1. Based on Zeng et al. research [Zeng GQ, Chen AB, Li W, Song JH, Gao CY. High MMP-1, MMP-2, and MMP-9 protein levels in osteoarthritis. Genet Mol Res. 2015 Nov 23;14(4):14811-22] and Tchetverikovet al. research [Tchetverikov I, Lohmander LS, Verzijl N, et al MMP protein and activity levels in synovial fluid from patients with joint injury, inflammatory arthritis, and osteoarthritis Annals of the Rheumatic Diseases 2005;64:694-698.], I think authors should compare their works with other studies and describe the significant found in this work.  
  2. Based on Hoff et al. research [Hoff P, Buttgereit F, Burmester GR, et al. Osteoarthritis synovial fluid activates pro-inflammatory cytokines in primary human chondrocytes. Int Orthop. 2013;37(1):145-151. doi:10.1007/s00264-012-1724-1], Hoff et al. investigated the increase of pro-inflammatory cytokines (including VEGF) for chondrocytes treated with OA synovial fluids. I think authors found the variation of MMP-1, MMP-13 and VEGF expressions in SW982 cells and THP1 cells treated with SFs (of culture media) of progressive OA grades. I think the results are interesting but need further researches, particularly in inflammatory mechanism .

Author Response

The authors thank the reviewer for his / her valuable suggestions. The manuscript is revised and improved as the suggestions. A detail response to the specific comments is attached herewith as a separate document. 

Reviewer 2 Report

The submitted manuscript entitled “Synovium-synovial fluid axis in osteoarthritis pathology: a key regulator of cartilage degradation process” by Dhanashri et al. reported that pro-inflammatory marker and MMPs levels in OA graded by KL score. Although, the manuscript contains some interesting observations, its overall novelty is limited. Because, these expression patterns have been reported in several studies (doi.org/10.1186/s12891-020-3120-0, doi.org/10.1111/sji.12770, doi: 10.1007/s00264-013-2192-y). In this manuscript, the authors didn't refer or discuss about such previous studies. Therefore, the authors must clarify novelty of the manuscript and address several concerns as bellow.

(1) When you want to mention high-low levels of markers in spite of no statistically significant difference, you should exhibit P-value and present tendency as P<0.1 at least. If P value is higher than 0.1, there is no difference.

(2) Overall, you should describe in result-paragraphs what expression level was analyzed such as “mRNA levels” or “Protein levels”. Especially, in figure 1 and 3, mRNA levels of several markers were analyzed? However, you showed concentration (pg/mL) in explanatory note of graph A, E, F and G. They should be removed.

(3) Graphs should be shown by box plot or scatter plot in Figure 1 and 3.

(4) MMP-2 and MMP9 mRNA in synovial biopsies also should be analyzed in Figure 1.

(5) The description for IL1β protein levels was should be shown in top of paragraph 3.2. because of Figure 2A.

(6) Table a1 and b1 seems not to match your description as shown in “MMP-1 up-regulation was significant between KL grade I and III (P<0.001) as well as KL grade I and IV (P<0.001). A significant difference in MMP-1 was also noted between KL grade II and III (P< 0.001) and KL grade II and grade IV (P<0.001)” and “The difference in the expression level was marginally significant between KL grade I and II (P<0.05) and KL grades I and III (P<0.05). Also, the expression variation between KL grade I and IV was noteworthy (P<0.01)”.

(7) Figure 4, 5 should place in main result and discuss about them.

(8) In figure 4, another inflammation marker such as TNF, IL1B and IL6 should be analyzed. Additionally, bar for SD must be shown in Figure 4B.

Author Response

(The authors gave the same response as above.)

Round 2

Reviewer 2 Report

Many points have been improved, but one problem remains.

All figures must be presented in order. To describe the results in Figure 3 and 4 alternatively is unkindness for readers. You should modify figure compositions. For example, Figure 3 as NO, IL1B and MMP expression in SW982, and Figure 4 as that in THP1.

Author Response

Response the reviewer 2 (round 2)

The reviewer’s comment: All figures must be presented in order. To describe the results in Figure 3 and 4 alternatively is unkindness for readers. You should modify figure compositions. For example, Figure 3 as NO, IL1B and MMP expression in SW982, and Figure 4 as that in THP1.

The authors’ response: The authors thank the reviewer for his / her suggestion to modify figures composition. As per the suggestion, both the cell lines data has been segregated now; Figure 3 describes all SW982 cell line data (inflammation induction – NO and IL-1β results as well as MMP specific response of the inflamed cells). Similarly, Figure 4 describes THP1 cell data. This figure modification will ensure swift reading of the manuscript.
